# Experiment level curation of transcriptional regulatory interactions in neurodevelopment

**Eric Ching-Pan Chu**[1,2,3], **Alexander Morin**[1,2,3], **Tak Hou Calvin Chang**[1], **Tue Nguyen**[1], **Yi-Cheng Tsai**[1], **Aman Sharma**[1], **Chao Chun Liu**[1], **Paul Pavlidis**[1,2]*

1 Michael Smith Laboratories, University of British Columbia, Vancouver, British Columbia, Canada,
2 Department of Psychiatry, University of British Columbia, Vancouver, British Columbia, Canada,
3 Graduate Program in Bioinformatics, University of British Columbia, Vancouver, British Columbia, Canada

* paul@msl.ubc.ca

**Data Availability Statement:** All relevant data are within the manuscript and its Supporting Information files. The code for all data analyses and figures generation are available on GitHub (https://

## Abstract

To facilitate the development of large-scale transcriptional regulatory networks (TRNs) that may enable in-silico analyses of disease mechanisms, a reliable catalogue of experimentally verified direct transcriptional regulatory interactions (DTRIs) is needed for training and validation. There has been a long history of using low-throughput experiments to validate single DTRIs. Therefore, we reason that a reliable set of DTRIs could be produced by curating the published literature for such evidence. In our survey of previous curation efforts, we identified the lack of details about the quantity and the types of experimental evidence to be a major gap, despite the theoretical importance of such details for the identification of bona fide DTRIs. We developed a curation protocol to inspect the published literature for support of DTRIs at the experiment level, focusing on genes important to the development of the mammalian nervous system. We sought to record three types of low-throughput experiments: Transcription factor (TF) perturbation, TF-DNA binding, and TF-reporter assays. Using this protocol, we examined a total of 1,310 papers to assemble a collection of 1,499 unique DTRIs, involving 251 TFs and 825 target genes, many of which were not reported in any other DTRI resource. The majority of DTRIs (965; 64%) were supported by two or more types of experimental evidence and 27% were supported by all three. Of the DTRIs with all three types of evidence, 170 had been tested using primary tissues or cells and 44 had been tested directly in the central nervous system. We used our resource to document research biases among reports towards a small number of well-studied TFs. To demonstrate a use case for this resource, we compared our curation to a previously published high-throughput perturbation screen and found significant enrichment of the curated targets among genes differentially expressed in the developing brain in response to Pax6 deletion. This study demonstrates a proof-of-concept for the assembly of a high resolution DTRI resource to support the development of large-scale TRNs.

## Author summary

The capacity to computationally reconstruct gene regulatory networks using large-scale biological data is currently limited by the absence of a high confidence set of one-to-one

github.com/PavlidisLab/dtri-curation). An additional copy of the data has been deposited in Scholars Portal Dataverse (https://doi.org/10.5683/SP2/XLLWRL).

**Funding:** This work was supported by National Institutes of Health grant MH111099 (https://www.nih.gov/) and Natural Sciences and Engineering Research Council of Canada grant RGPIN-2016-05991 (https://www.nserc-crsng.gc.ca/), both held by PP. The funders had no role in study design, data collection and analysis, decision to publish, or preparation of the manuscript.

**Competing interests:** The authors have declared that no competing interests exist.

regulatory interactions. Given the lengthy history of using small scale experimental assays to investigate individual interactions, we reason that a reliable collection of gene regulatory interactions could be compiled by systematically inspecting the published literature. To this end, we developed a curation protocol to examine and record evidence of regulatory interactions at the individual experiment level. Focusing on the area of brain development, we applied our pipeline to 1,310 publications. We identified 3,601 individual experiments, providing detailed information about 1,499 regulatory interactions. Many of these interactions have verified activity specifically in the embryonic brain. By capturing reports of regulatory interactions at this level of detail, we equip the users with more granular information than other similar resources, enabling more informed assessments of reliability.

## Introduction

Reconstruction of transcriptional regulatory networks (TRNs) has the potential to enable in-silico analysis of developmental processes and disease mechanisms. As such, using high-throughput biological data to infer large scale TRNs is an area under active research; recent examples include [1–4]. However, the utility of these TRNs has been hindered by the absence of a high confidence set of regulatory interactions for training and validation. Researchers have historically used less scalable experimental techniques to investigate direct transcriptional regulatory interactions (DTRIs). While low-throughput, such methods tend to be considered reliable, especially if there are multiple independent lines of evidence supporting a DTRI. Thus, there would be value in having resources that aggregate high-quality reports of DTRIs, forming the topic of the current work. Our particular interest is in DTRIs of relevance to the developing nervous system, as mutations in transcription factor (TF) genes [5–7] and regulatory regions [8–10] have been highly implicated in neurodevelopmental disorders.

We define DTRIs as pairwise interactions between a transcription factor (TF) and a target gene where the TF modulates target expression by associating with a cis-regulatory element (cRE). In the interest of inclusivity, we are defining direct regulation to broadly include both scenarios where the TF binds directly to DNA via its DNA binding domain or by interacting with other TFs. This definition of "directness" is in contrast with "indirect regulation" where two or more DTRIs act in series in a pathway. There are three types of low-throughput experimental paradigms commonly used to elucidate DTRIs, including TF perturbation, TF-DNA binding, and TF-reporter assays (Fig 1A). In TF perturbation assays, manipulation of TF expression is followed by an assessment of target gene expression. In TF-DNA binding assays, protein-DNA interactions between the TF and the cRE are evaluated. Finally, TF-reporter assays measure the functional impact of the TF binding on the associated cRE sequence. While low-throughput assays are not infallible, they generally yield higher confidence than high-throughput alternatives by evading the need for large scale inferential statistics and enabling detailed and readily replicable characterization of single DTRIs; examples: [11,12] (Fig 1B). Notably, such low-throughput experiments are routinely used to validate putative targets identified by more scalable approaches. Given the importance and wide acceptance of these types of evidence, it would be useful to assemble a centralized catalogue of DTRIs that is supported by low-throughput experimental evidence in the published literature.

There have been a number of earlier efforts to aggregate DTRIs from the literature: ENdb: [13], TRRUST: [14,15], CytReg: [16], OReganno: [17–19], HTRIdb: [20], TFe: [21], TFactS: [22], InnateDB: [23]. None of these curation efforts were tailored to the neurodevelopment

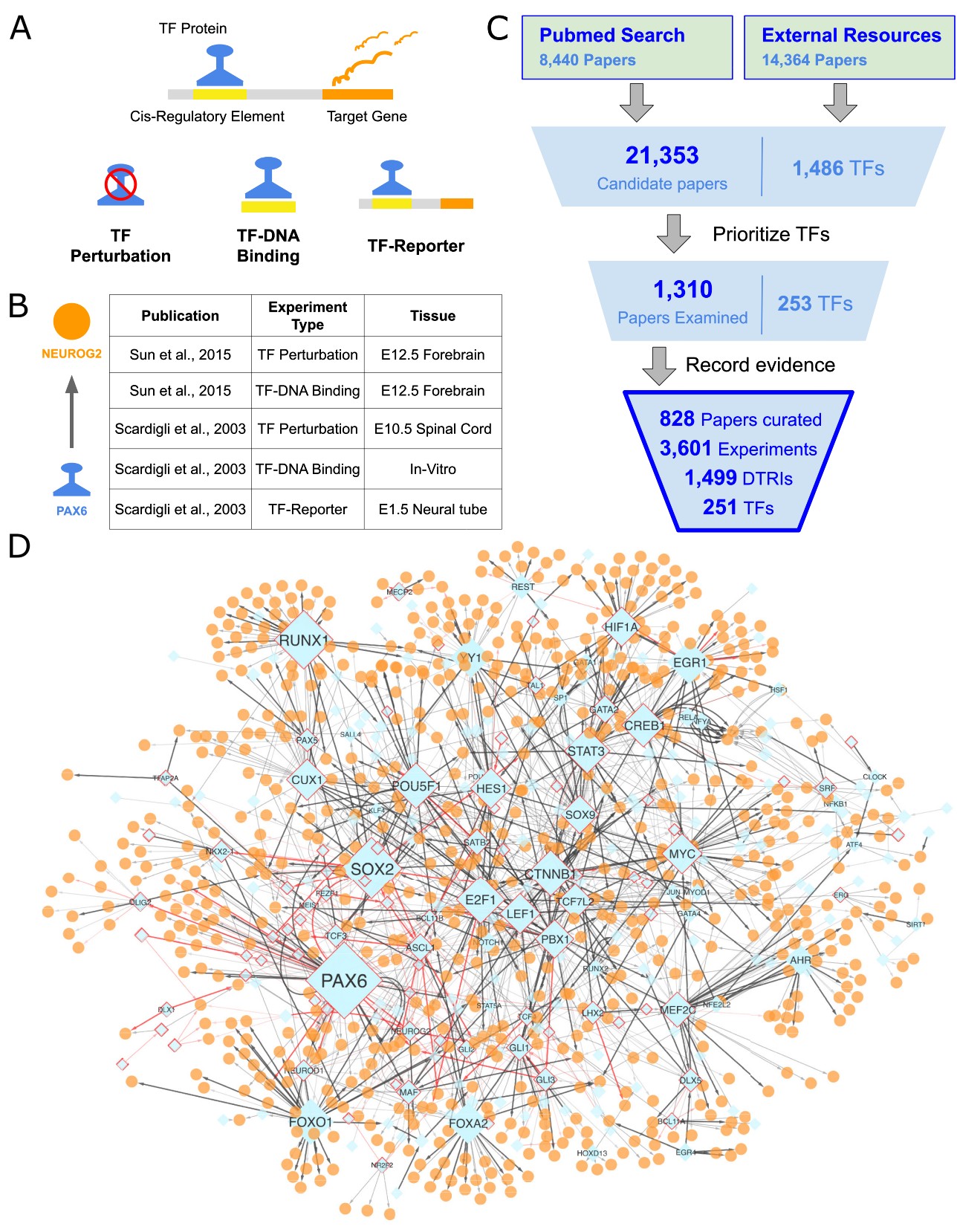

**Fig 1. Summary of curation.** (A) A schematic of the three types of low-throughput experimental evidence we considered. (B) Curated records for the interaction between PAX6 and NEUROG2 provided as an example. (C) Overall curation workflow. The count of papers and the corresponding number of TFs in each stage of curation are printed. Summary statistics of the final curation output are displayed at the bottom (D) The manually curated regulatory network. TFs are diamond shaped and colored blue. Targets are circles and colored orange. The sizes of TFs correspond to the number of targets. TFs with 10 or more targets are labeled with the official HGNC gene symbol. Edge transparency corresponds to the number of types of experiments. Red edges represent DTRIs with experimental validation in primary CNS tissues or cells. Only the largest network component with 946 nodes and 1481 edges is displayed.

context. More importantly, these studies generally record annotations of pairs of interacting genes but capture little information about the underlying experimental evidence. This is notable because the type and quantity of evidence is expected to affect the reliability of a reported interaction. Specifically, each individual type of evidence provides only a limited view of any given DTRI. TF perturbation assays enable the assessment of the TF's ability to modulate target gene expression but cannot decipher its functional dependence on physical binding. Likewise, while TF binding at a cRE is necessary for regulation, detection of TF-DNA binding alone is insufficient for demonstrating functional activity. TF-reporter assays simultaneously demonstrate both functional modulation and physical binding but often by examining the given DTRI outside of the native genomic and cellular context. As such, integration across these types of experiments should help establish DTRIs with high confidence.

Driven by the premise that curation of details at the individual experiment level would facilitate accurate evaluation of reliability, we undertook a systematic effort to inspect the literature in this way. Consequently, the resulting resource contains more granular information about the experimental evidence underlying each report of DTRI than other similar resources, enabling users to make more informed and independent assessment of confidence. Finally, our curation effort provides a partial summary snapshot of the literature landscape surrounding transcriptional regulation in the developing brain.

## Results

### Overview of curation

Our curation pipeline is summarized in Fig 1C (see S1 Appendix for details). Briefly, for each TF, we assembled a set of candidate papers (S1 Data). Next, we manually prioritized TFs for curation based on annotated associations with central nervous system (CNS) development and the number of candidate papers retrieved (S2 Data). For each paper examined, we recorded the details of all reported experiments that lend support to any DTRI in humans or mice (Box 1 and S3 Data). For reporting, we mapped all genes to human orthologs while retaining the species information as an additional feature. Applying this pipeline to a total of 1,310 papers, we established a collection of 1,499 unique DTRIs, involving 251 TFs and 825 targets, from 828 papers. This manually curated network is displayed in Fig 1D and the complete set of curated interactions are provided in S4 Data. In the following sections, we present a detailed summary of the curated data resource and compare it to a high-throughput TF perturbation screen.

### Identification of candidate papers highlights biases in TF coverage

The input to our curation was a corpus of candidate publications. To establish this corpus, we started by taking advantage of previous curation efforts. We obtained 14,364 papers from seven external resources, covering 1,305 TFs (Fig 2A and S1 Data). TRRUST, the largest database of literature curated DTRIs, provided more than 10,000 publications but only recovered about ~30% of those recorded in the other resources (Fig 2B and 2C). Further, overlaps among

## Box 1. Experimental details recorded during curation

| | |
|---|---|
| Experiment Type | One of three types of experiment being curated.<br>*Value*: *TF Perturbation*, *TF-DNA Binding*, or *TF-Reporter*. |
| Context Type | A broad classification of the cellular context tested.<br>*Value*: *Primary Tissue*, *Primary Cells*, *Cell Line*, or *In-Vitro*. |
| Cell Type | An ontology term that best corresponds to the tissue or cell type used.<br>*Example*: *UBERON:0001017 (central nervous system)* |
| TF Species | The species of the TF protein or sequence.<br>*Value*: *Human or Mouse*. |
| Target Species | The species of the target protein or regulatory element.<br>*Value*: *Human or Mouse*. |
| TFBS Position | A broad classification of the distance between the transcription factor binding site (TFBS) and the target transcription start site (TSS).<br>*Value*: *Proximal or Distal*. |
| Mode | The mode or direction of regulation.<br>*Value*: *Activation or Repression*. |
| [1]Details | TF Perturbation<br>Effect: *Knock Out*, *Knock Down* or *Overexpression*<br>[2]Type: *Constitutive or Induced*<br>TF-DNA Binding<br>Method: *ChIP-assay* or *EMSA (Electrophoretic Mobility Shift Assay)*<br>[3]TF Source Type: *Primary Tissue*, *Primary Cells*, or *Cell Line*<br>TF-Reporter<br>Mutated: *TRUE or FALSE*<br>[4]Binding Verified: *Putative*, *EMSA*, or *None* |

[1]Different sets of details are recorded for different types of experiments. [2]Constitutive perturbation refers to mutations that are present throughout the course of development, as opposed to induced perturbations using Cre-loxP or RNA interference that are triggered closer to the time of assay. [3]The TF protein used in EMSA experiments may be sourced differently across experiments. In many cases, it is obtained from cell lines after TF transfection. In other cases, nuclear extracts containing the endogenous protein are obtained from primary tissues or cells. [4]For TF-reporter assays where the cRE sequence is mutated and tested, the impact of the mutation on TF-DNA binding is sometimes verified using additional EMSA experiments.

the other resources are generally small (0%-22%). These observations suggest that there may be additional papers in the literature containing reports of DTRIs. As such, we expanded the pool of candidate papers by searching PubMed using several relevant Medical Subject Headings (MeSH) terms (see Methods for details). We identified an additional set of 6,989 candidate papers for 1,140 TFs (Fig 2A). In particular, for TFs directly associated with CNS development, we were able to increase the total number of candidate papers from 5,729 to 9,839. Together, we assembled a set of 21,353 candidate papers covering 1,486 TFs.

We assessed coverage of TFs among the retrieved set of candidate papers. We found that 90% of all the publications recorded in previous curation databases were covered by the top 316 TFs. Similarly, 90% of all candidate papers queried from PubMed were covered by the top 289 TFs. The set of top TFs in previous curation databases overlaps substantially with the set of top TFs identified in our independent PubMed query (Jaccard Index = 0.52), suggesting shared biases. The overall pattern is shown in Fig 2D. Further, a substantial fraction of TFs (749; 34%) had no candidate papers. Importantly, some key neurodevelopmental TFs appear to have had very limited investigation. For example, TBR1 is a TF recently implicated in Intellectual Disability (ID) and Autism Spectrum Disorder (ASD) [24]. Despite this, we were able

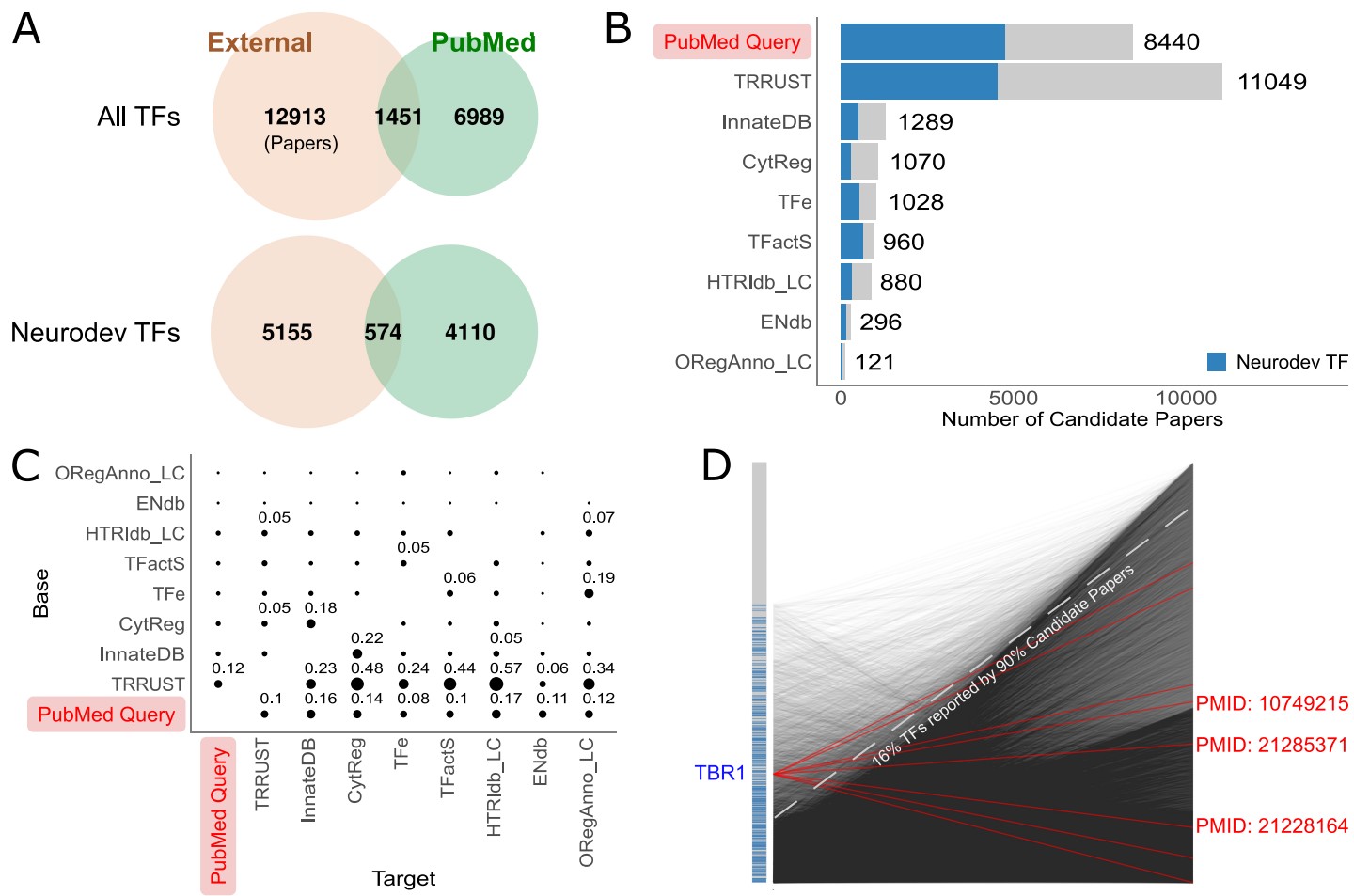

**Fig 2. Overview of the candidate literature corpus.** (A) Venn diagrams showing the overlap between candidate papers sourced from external data resources versus our independent PubMed query, broken down by all TFs (top) and those associated with neurodevelopmental TFs (bottom) (B) Number of candidate papers retrieved from each data resource. Papers associated with neurodevelopmental TFs are highlighted in blue. (C) Pairwise overlap of candidate papers among the different sources shown as fractions of the "target" source (horizontal axis). For example, TRRUST contains 0.48 of the papers recorded in CytReg whereas CytReg contains only 0.05 of the papers in TRRUST. Only values of 0.05 or higher are displayed. External resources are ordered by the number of recorded publications. (D) Summary of associations between TFs and candidate papers. Each point on the left vertical axis is a TF, ordered by the number of candidate papers assigned (TFs with the highest number of candidate papers are at the bottom). Neurodevelopmental TFs are highlighted in blue. Each point on the right vertical axis is a candidate paper, ordered by the number of associated TF (papers with the highest number of candidate papers are at the bottom). Each line denotes a TF-paper association. For example, TBR1 (highlighted) is associated with eight candidate papers (red lines), the PubMed IDs of three papers are printed as examples.

to identify only eight candidate papers for this gene (Fig 2D), suggesting that TBR1 was not previously popular enough to warrant much attention. We hypothesized that this bias in TF coverage reflects gene popularity differences in general. As expected, we found that the total number of papers per TF in PubMed is highly correlated with the number of candidate papers retrieved (Spearman's correlation = 0.86). As we discuss later, these biases in the literature influence the resulting database of interactions and its interpretation.

## Identification of 1,499 experimentally verified TF-target interactions

In total, we recorded 3,601 experiments by examining 1,310 research publications, providing high resolution evidence for 1,499 unique DTRIs involving 251 TFs and 825 targets (Fig 3A

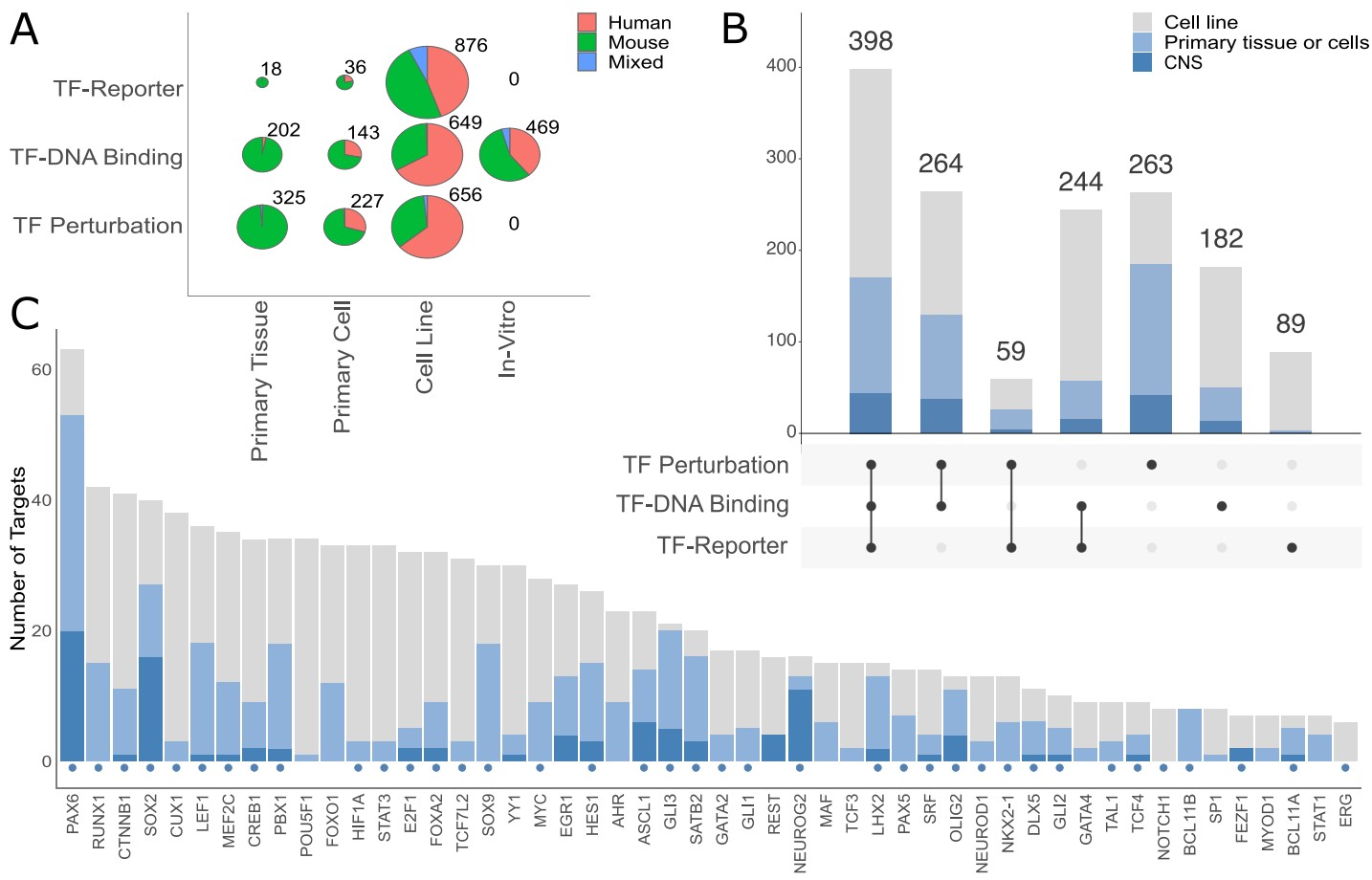

**Fig 3. Summary of recorded DTRIs.** (A) Breakdown of recorded experiments across experiment types and context types. The "in-vitro" category only includes EMSA experiments for testing TF-DNA binding. (B) Breakdown of DTRIs across different combinations of experiment types. Colors correspond to cellular contexts indicated in the color legend. (C) Number of targets per TF for the top 50 TFs. Color coding is the same as in (B). Neurodevelopmental TFs are indicated by blue circles. There are 39 TFs with more than ten targets (denoted by the horizontal dotted line).

and S3 Data). A small fraction (204; 14%) of all DTRIs were supported by evidence in both humans and mice (Fig 3A). About half (798; 53%) were reported only for mice and the remainder (497; 33%) only for humans. We were able to annotate 39 TFs with 10 or more DTRIs (Fig 3C). Collectively, these top 39 TFs regulate more than half (1018; 68%) of all curated DTRIs. The remaining 481 (32%) DTRIs were distributed across 212 TFs (S1 Fig). Unsurprisingly given our TF selection criteria, 31 of the top 39 TFs, are associated with neurodevelopment (Fig 3C). Notably, PAX6, a key TF implicated in corticogenesis [25,26] has 63 recorded targets. Further, we identified 12 targets with ten or more recorded TF regulators (S2 Fig). Eight of these 12 targets are themselves neurodevelopmental TFs including HES1, ASCL1, NEUROG2, MEF2C among others (S3 Fig). In particular, HES1, a TF known to be involved in the proliferation of neural progenitors [27], has 14 experimentally verified TF regulators.

Next, we looked at the overall patterns of experimental evidence underlying the recorded DTRIs. TF-DNA binding experiments were most common (1,463), followed by 1,208 TF perturbation and 930 TF-reporter assays (Fig 3A). The majority of all three types of experiments have been performed using immortalized cell lines (2,181 experiments) rather than primary tissues or cells (951 experiments). This was especially true for reporter assays where close to 95% (876/930) of the experiments were performed using cell lines (Fig 3A). Despite the overall

small number of experiments using primary tissue or cells, close to half of all DTRIs (620; 41%) were validated using at least one such experiment. Further, we found that the majority (965; 64%) of DTRIs were supported by two or more types of evidence and 398 (27%) DTRIs were supported by all three (Fig 3B). Of the DTRIs with all three types of evidence, 170 had been tested using primary tissues or cells and 44 had been tested directly in the CNS (Fig 3B).

For each type of experiment, we further explored a number of factors that may influence reliability of the reported DTRI (Box 1). For instance, the majority (845; 70%) of TF perturbation experiments recorded were performed by knocking down or knocking out TF expression, as opposed to inducing TF overexpression (363; 30%) (S4 Fig and S5 Data). Further, in TF perturbation experiments that use primary tissues or cells, we found that it is common (373 experiments) to induce a constitutive loss-of-function mutation in the TF and then compare the resulting target gene expression to that of wildtype samples (S4 Fig). The presence or absence of genetic manipulation throughout the organism's development introduces an additional layer of complexity for interpretation. Of the 550 TF perturbation experiments performed using primary tissues or cells, 177 (32%) induced the perturbation closer to the time of assay by employing strategies such as Cre-LoxP or RNA interference. For TF-DNA binding experiments, we found that the majority of experiments (994; 68%) used chromatin immunoprecipitation to test for in-vivo binding events in either primary samples (345 experiments) or immortalized cell lines (649 experiments) (S5 Fig and S6 Data). EMSAs were also commonly employed to test for in-vitro TF protein-DNA interactions (469 experiments). Further, we found a small number of EMSA experiments (48) that used proteins obtained by nuclear extractions from primary tissues or cells (S5 Fig and S6 Data). Finally, for TF-reporter experiments, we recorded whether mutated versions of the TFBS sequence were assayed to confirm a direct binding mechanism. We found that 407 of the 930 reporter gene assays examined the functional consequence of mutating the corresponding TFBS sequence (S6 Fig and S7 Data). Overall, the granularity of our curation highlighted a wide range in the quality and quantity of evidence supporting the reported DTRIs.

Our curation also accounted for tissues and cell types, which we recorded at the highest resolution possible with existing ontologies. This allows subdivision of the data in terms of relevance to particular contexts. In total, 951 (26%) experiments recorded (for 620 DTRIs) were performed using primary tissues or cells. In terms of anatomical systems, among these experiments, the most represented was the CNS, with 243 experiments (155 DTRIs) (S7 Fig). The set of DTRIs in the CNS is highly enriched for neurodevelopmental TFs (p-value $< 5.6 \times 10^{-9}$, hypergeometric test). Further, a large fraction (181; 74%) of these experiments used embryonic CNS samples, thus providing evidence of activity in the developing CNS (S7 Fig). For example, ASCL1, FGF19, and SOX2 were reported to regulate targets in the embryonic telencephalon [28], diencephalon [29], and neural stem cells [30], respectively. We also found some DTRIs involving known neurodevelopmental TFs that were assayed only in other tissues, such as a small number of PAX6 targets in pancreatic islets [31,32] and small and large intestine [33]. Over half (2,181; 60%) of all experiments were performed in cell lines, regardless of the experiment type (Fig 3A). Among these, the most popular were kidney derived cell lines (S8 Fig). As expected, cell line experiments accounted for a larger proportion of human samples compared to primary tissue or cells (Fig 3A). Our detailed information about cellular contexts allows efficient and accurate data subdivision based on user requirements.

Next, we assessed overlaps with other DTRI resources. Since we sourced many candidate papers directly from such earlier curation efforts, a significant amount of overlap is expected. By examining 657 previously curated papers, we managed to extract 809 DTRIs from 467 papers but failed to identify low-throughput experimental evidence in the remaining 190 papers (Fig 4A and S1 Data). At the level of DTRIs, 40% of our database overlaps with

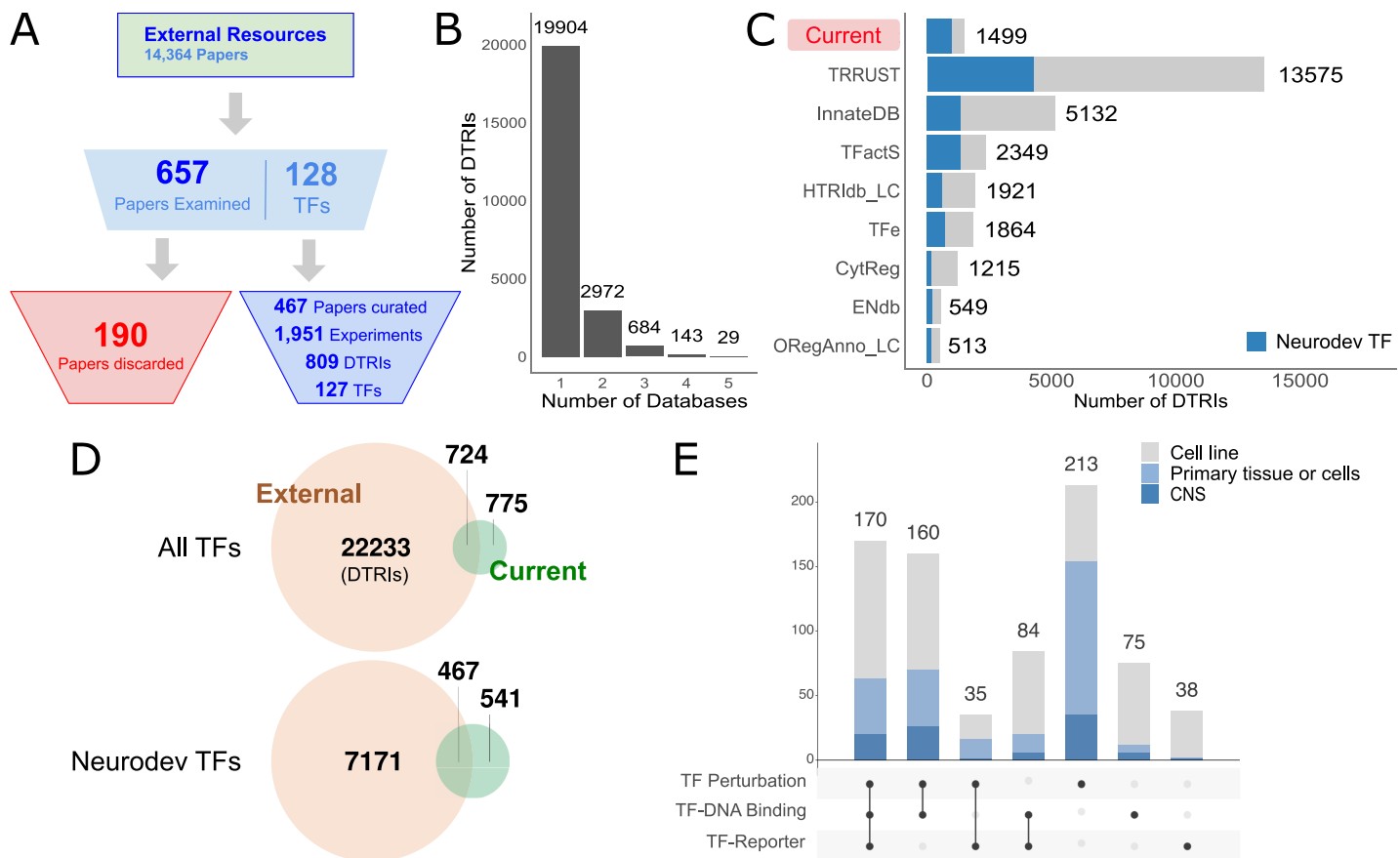

**Fig 4. Comparison with external resources.** (A) Flow chart of candidate papers obtained from seven external resources. (B) Distribution of DTRIs across the number of resources. Most DTRIs are only recorded in a single database whereas 29 DTRIs are recorded in five databases. No DTRIs have been recorded by more than five databases. (C) Number of DTRIs in each database. DTRIs involving neurodevelopmental TF regulators are highlighted in blue. (D) Venn diagrams showing the overlap between DTRIs recorded in previous curation databases versus the current study, broken down by all TFs (top) and those associated with neurodevelopmental TFs (bottom). (E) Breakdown of DTRIs unique to our curation across different combinations of experiment types. Colors correspond to cellular contexts indicated in the color legend.

TRRUST while other resources contain up to 8% of our records (S9 Fig). Limited overlap is common among the other resources as well, with the overwhelming majority of DTRIs having been recorded only in a single database (Fig 4B). This demonstrates the general incomplete coverage of the literature even by the most comprehensive curation efforts to date. Despite being smaller than most other resources (Fig 4C), we still managed to identify 775 DTRIs that were not previously curated in any other database, 541 of which directly involved a neurodevelopmental TF (Fig 4D). Importantly, 449 (58%) of these newly identified interactions were supported by multiple lines of experimental evidence (Fig 4E). Taken together, our curation has expanded the repertoire of annotated DTRIs among the existing DTRI data resources.

Because we curated only a fraction of the literature, it is of interest to estimate the total number of DTRI reports with low-throughput experimental evidence in the remainder. We base our estimate on the observation that of the 1,310 candidate papers that we examined, 63% (828) were found to contain at least one report of DTRI. It follows that approximately >12,000 of the remaining 20,043 candidate papers contain experimental evidence of DTRI. With an average of 1.9 DTRIs reported by any single publication (S10 Fig), there would be >22,000 DTRIs remaining in the literature, in addition to the 1,499 DTRIs already curated. Limiting to

the 8,730 candidate papers annotated with neurodevelopmental TFs, there would be >10,000 more DTRIs. As noted above, we found that under a third (398/1,499) of all recorded DTRIs were supported by all three types of experimental evidence. Further, about half (170) of these high confidence DTRIs had been tested in primary tissues or cells, and of these, 44 were tested directly in the CNS. Assuming the set of candidate papers is representative, there would be around >6,000 (0.3*22,000) more DTRIs that are supported by all three types of evidence and >600 (0.03*22,000) of these would have been tested using primary CNS tissues or cells. By extension, we estimate our curation of 44 DTRIs with high confidence verification of activity in the CNS captures up to 7% of all such DTRIs reported in the literature. Taken together, we estimate that there remain many thousands of additional high-confidence DTRIs in the published literature which could be the subject of future curation efforts.

### Network properties reflect potential research biases

Given the literature biases in coverage of TFs (Fig 2D), we suspected that similar biases may exist in the selection of regulatory targets. Specifically, researchers may be more likely to choose to investigate interactions between genes that are suspected to be related. If this is true, the manually curated network should be more connected than expected if the targets were chosen randomly. To test this, we integrated all DTRIs in our database to construct a directed network consisting of 955 nodes and 1,499 edges (Fig 1D). We measured network connectivity in three ways. First, we counted the number of valid gene-to-gene paths in the network. Briefly, for every gene in the network, we counted the number of other genes that are within reach via at least one continuous path. The total count was then obtained by summing across all genes. Because the edges are directed and the network consists of multiple components, not every gene is reachable from every other gene in the network. In total, we observed more than 77,000 gene-to-gene paths in the curated network, which is significantly higher than the mean of 55,411 paths among a null constructed from random networks (p-value < 0.01; see Methods). This indicates a high degree of global connectivity within the network. Next, we counted the number of cliques with three or more nodes, ignoring directionality. We found 215 cliques in the curated network, which is higher than a mean of 140 cliques among the random networks (p-value < 0.01), demonstrating a large number of locally interconnected modules. Finally, we observed only four independent components in the curated network whereas a typical random network had 26 components (p-value < 0.01), implying that even peripheral genes with low node degrees remain connected to the rest of the network. Taken together, the manually curated network is highly interconnected, even after controlling for biases in TF coverage. This strongly suggests substantial biases in the selection of targets by investigators, as observed for TF selection.

Continuing our investigation of biases in the data, we hypothesized that TSS proximal cREs would be enriched among the reported DTRIs since distal elements are likely more difficult to identify. We define proximal regulatory elements to be either promoters or regulatory elements that fall within 2 kb of the target TSS, as indicated by the original publication. The 2 kb threshold was previously used by ENCODE to classify proximal vs. distal enhancers [34]. We found that most (604 of 663 DTRIs where the TFBS position was annotated) of the reported DTRIs involve proximal cREs and only 74 DTRIs have been annotated with distal regulatory sites (S11 and S12 Figs; this observation is independent of the 2 kb threshold that we chose; S13 Fig). Distal sites include the interaction between SOX2 and SHH where the corresponding enhancers are >5 kb away from the TSS [35]. Such cases are, by far, the minority in our curation. In addition to TFBS proximity, we also annotated whether a regulatory interaction is activating or repressive, referred to as the mode of regulation (Box 1). We found that about less

than a third (313/1,317) of the DTRIs are repressive (S11 Fig). It is less clear whether this trend reflects underlying biological trends or another form of investigator bias in the selection of interactions to study. Notably, several repressive DTRIs involve TFs that are generally characterized as repressors including HES1, GLI3, and REST (S14 Fig). In particular, HES1 was annotated to repress 16 of its 26 targets (among the 23 targets where the mode of regulation was reported). The majority of other TFs have been found to upregulate the expression of the majority of their target genes. For example, RUNX1 represents a typical example that activates expression of 80% (30/37) of its recorded targets. A small number (23) of DTRIs have been reported to be both activating and repressive in different experiments. Overall, of all 581 DTRIs where both the mode of regulation and the cRE positions are known, 416 (72%) involve activation of proximal cREs and only 10 (<2%) are repression of a distal cRE.

## Comparison with high throughput TF perturbation screens

One application of our curated DTRI resource is to benchmark high-throughput screens. To demonstrate this use case, we analyzed a previously published TF perturbation screen for Pax6 (Fig 5A). In this study, the authors sought to identify Pax6 targets in the embryonic mouse forebrain by examining genome wide differential expression between wildtype and Pax6 mutant mice using microarrays [36]. We assessed enrichment of our curated PAX6/Pax6 targets in this dataset (this includes targets validated in either humans or mice). We found that 22 of all 56 curated PAX6/Pax6 targets were differentially expressed at a false discovery rate (FDR) of 0.1 (p-value $< 4.4 \times 10^{-6}$, hypergeometric test; similar results were obtained with a threshold-free comparison (S15 Fig)). Among these include several known neurodevelopmental genes such as ASCL1, SOX2, and NEUROG2 (Fig 5A). We conclude there is significant correspondence between the curated targets and the high-throughput differential expression screen.

In the analysis above, we ignored context, but it is plausible that low-throughput experiments performed in the same tissue as Walcher et al., 2013 [36] would yield even higher

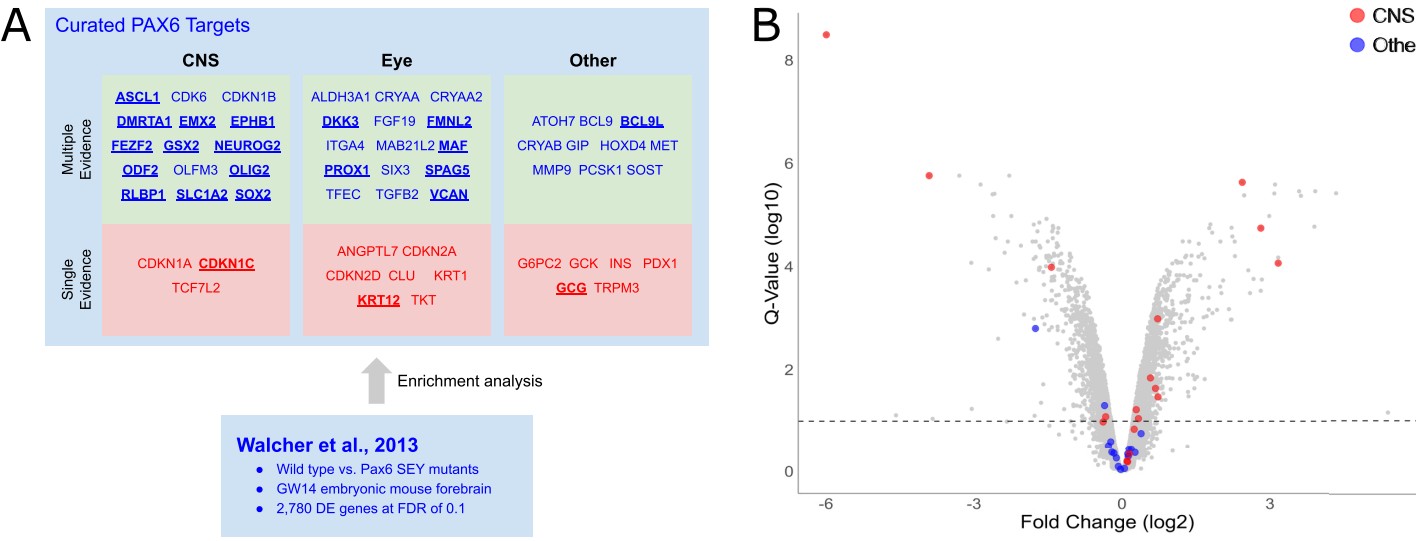

**Fig 5. Comparison with a high-throughput Pax6 perturbation screen.** (A) Summarized result of the comparative analysis. Curated PAX6/Pax6 targets are organized by tissue and number of types of experimental evidence. Genes differentially expressed at FDR of 0.1 are bolded and underlined. (B) Volcano plot of differential expressions between wildtype and Pax6-Sey mouse embryonic forebrains. Curated PAX6/pax6 targets are highlighted with tissue specificity as depicted in the color legend. Fold change ($\log_2$) is displayed in x-axis. Multiple test corrected q-values (Benjamini-Hochberg Procedure) are plotted on the y-axis on a log10 scale. Dotted line indicates 0.1 FDR for reference. Curated CNS targets are highly overrepresented in the differentially expression profile (p-value $< 8.5 \times 10^{-9}$, hypergeometric test) whereas no overrepresentation was detected for the targets in the "Other" category (p-value $< 0.47$, hypergeometric test).

correspondence. To test for this, we divided the PAX6/Pax6 targets into three tissue types: the CNS, the eye, and "other", with the latter containing mostly DTRIs validated in cell lines. Since the differential expression profile was generated in the embryonic mouse forebrain, we hypothesized that the targets supported by low-throughput CNS evidence would be most highly enriched. We found that this is indeed the case. Thirteen of 18 CNS targets were differentially expressed at an FDR of 0.1 (p-value $< 8.5 \times 10^{-9}$, hypergeometric test) (Fig 5B). This is nearly a twofold improvement over the set of all curated PAX6/Pax6 targets and six-fold over those in the "other" category where only two of 16 targets were differentially expressed (p-value $< 0.47$, hypergeometric test) (Fig 5A and 5B). Again, this observation was corroborated by an additional threshold-free analysis (S16 Fig). Further, we confirmed that the increase in the level enrichment for CNS targets over the set of all PAX6/Pax6 targets and those in the "other" category were statistically significant by using a resampling strategy to estimate the $95^{th}$ percentile confidence intervals of the enrichment values (S17 Fig). Using the same strategy, we found that the CNS targets were also significantly more enriched than the targets recorded in TRRUST, which expectedly had a similar level of enrichment as the set of all PAX6/Pax6 targets in our resource (S17 Fig). Finally, the level of enrichment for targets validated in the eye is approximately the same as the set of all targets (7 of 22 targets were differentially expressed at FDR of 0.1; p-value $< 1.5 \times 10^{-2}$, hypergeometric test). Similar to cellular contexts, the level of enrichment was higher for the targets with multiple types of recorded experiments (19 of 40 targets differentially expressed at FDR of 0.1; p-value $= 4.2 \times 10^{-7}$, hypergeometric test) than those with only a single type (3 of 16 targets differentially expressed at FDR of 0.1; p-value $= 0.24$) (Figs 5A and S18). We were able to replicate these general patterns using an alternative Pax6 perturbation dataset that was generated using the embryonic mouse cortex [37], though the differences did not pass the threshold for statistical significance (S19 and S20 Figs). This is consistent with possible differences in data quality among high-throughput datasets. Taken together, this analysis demonstrates the utility of having detailed experimental and contextual information for the evaluation of high-throughput screens.

Next, we asked whether the level of correspondence generalizes to TFs other than PAX6/Pax6. To do this, we sourced 53 high-throughput TF perturbation datasets for 10 TFs from KnockTF [38]. We only considered TFs with 3 or more TF perturbation datasets. KnockTF contains a heterogenous set of tissue and cell types including both primary tissue or cells and immortalized cell lines. To reduce noise and control for differences in conditions among the individual datasets, we generated consensus p-value rankings for each TF by averaging across all corresponding datasets. This allowed us to compute a single AUROC enrichment value for the curated targets in each TF. In eight out of the 10 TFs, the level of enrichment improved in the consensus ranking over the median level of enrichment across the individual datasets, confirming the validity of the aggregation approach (S21 Fig). Importantly, the level of enrichment reached statistical significance for 4 of the 10 TFs, corroborating signals of correspondence between the curated targets and high-throughput TF perturbation screens beyond PAX6/Pax6. The level of enrichment is notable given the sparse coverage of our curation and the expected level of noise in the high-throughput datasets. To compare with other DTRI resources, we obtained an additional set of 123 TF perturbation datasets for 27 other TFs. This allowed us to compute AUROC enrichment values for all DTRI resource and TF combinations where there are 10 or more targets. Note that this analysis is constrained by the limited coverage of targets among the DTRI resources. For example, ENdb has 10 or more targets in the consensus p-value ranking of only a single TF (FOXA1) among all the TFs considered (S13 Data). Based on this limited set of comparisons, we found that the level of correspondence with high-throughput perturbation data is generally comparable among the various DTRI resources (S22 Fig).

## Discussion

The elucidation of the genetic circuits underpinning neurodevelopmental disorders has been a major challenge. While there has been progress in the development of TRN reconstruction methods using high-throughput data, it is reasonable to ask how much has already been captured in the lengthy history of low-throughput experiments, and to make maximal use of this information. Because low-throughput methods appear to be considered reliable (they are often used to validate high-throughput methods), especially when there are multiple lines of evidence, having a high-quality assembly of such data would be beneficial. In our survey of previous efforts to produce such resource, we identified the lack of detail about the amount and type of low-throughput evidence to be a major gap. To this end, we undertook a systematic and detailed effort to inspect the published literature for support of DTRIs at the individual experiment level. We demonstrated the utility of the available annotations in our resource by comparing with high-throughput perturbation datasets. Here, we release the result of our curation for use by the wider research community.

The significance of having experiment level details is emphasized by our observation of the variation in confidence levels across the reported DTRIs in the published literature. A summary analysis of our database has revealed that only a fraction of all reported DTRIs is supported by all three types of experimental evidence. Similarly, most experiments were carried out using cell lines that may not recapitulate in-vivo activity. Even within a single type of experiment, we observed important differences in approach that may further impact reliability. This pattern of varied confidence among the reported DTRIs was largely unreported in previous curation efforts. The practical utility of our curation is further demonstrated by the comparison with the high-throughput Pax6 perturbation screen. The significantly higher levels of enrichment among the targets verified directly in the CNS and by multiple evidence types further confirms that our curation approach is appropriate for compiling bona fide DTRIs in the low-throughput literature.

Among all the external resources reviewed, CytReg [16] and ENdb [13] were the only studies with explicit consideration of different types of experimental evidence. In CytReg, the authors classified experiments into functional or binding assays, abstracting away other details that may further impact reliability. In ENdb, the authors focused on the annotation of enhancers, with limited curation of details on specific TF-target interactions. TFe, on the other hand, took a more implicit approach to account for reliability [21]. Instead of using a systematic curation approach, TFe relied on annotations of TF-target interactions by community experts, who in turn evaluate the merits of individual studies on an ad hoc basis. A recent report provided evidence to support the quality of the DTRIs recorded in TFe [39]. However, the fractured curation process makes this approach have questionable scalability. TRRUST is the largest literature-based resource to date [14,15]. Using a semi-automated text mining approach, TRRUST recorded pairs of interacting genes from publication abstracts, purposely overlooking the underlying experimental details. The remaining databases including OReganno [17–19], HTRIdb [20], TFactS [22], and InnateDB [23] rely on single experiments, which is mostly commonly the TF-DNA binding assay, to indicate direct regulation. Importantly, many of these resources included high-throughput evidence, either purposely or unintentionally. Finally, except for InnateDB and ENdb, none of the previous curation efforts recorded information about the cellular context.

There are still a number of limitations to our work. In general, manual curation can have errors. In order to minimize mistakes, we established and strictly followed a formal curation protocol (S1 Appendix). In particular, we introduced controlled vocabularies for all recorded attributes to simplify the curation process. All records were checked twice, and any conflicts

were resolved by the first author. Next, incomplete retrieval of candidate papers is a potential concern. While we strived to find as many papers as possible using both previous curation resources and independent PubMed queries, it is plausible that we have missed some candidate papers given our selection of and reliance on the MeSH search terms. Nonetheless, the pronounced popularity biases we report are unlikely to be an artifact of our search strategy. Further, since we aimed to curate only a handful of DTRIs for a small set of TFs of interest, an incomplete pool of candidate papers was not a major issue. However, for a more comprehensive curation effort with the goal of increasing coverage of less popular TFs, it is possible that a future study may benefit from using more elaborate text mining approaches for retrieving candidate papers.

In order to establish a direct binding mechanism for regulatory interactions with the highest possible confidence, the effect of modifying cREs in their endogenous chromosomal loci should be considered. Emerging studies are using CRISPR-KRAB and related approaches to perform this analysis; recent examples include [40,41]. However, such studies are few, and therefore have not been included in our curation protocol. Instead, we focused only on the three most commonly reported types of experimental evidence. In the future, it may be possible to integrate such data types in order to improve reliability beyond current standards.

The lack of a negative set may limit the utility of this resource for validation. For example, in the PAX6/Pax6 analysis we could only assess sensitivity of the high-throughput perturbation study with respect to our database, not specificity. This is because our attempts to find negative examples was largely unsuccessful. During curation, we took note of any TF perturbation or TF-reporter experiments that yielded negative results in the papers that we examined. We only found 11 such cases (S11 Data). There are two possible reasons for this. First, our search for candidate papers may be biased against experiments with negative results. If this is the case, it may be possible to improve the search strategy to identify more relevant papers. However, it is more plausible that the literature itself is depleted of negative reports, due to "the file-drawer problem" [42]. Future work should attempt to identify negative examples by performing further experimentation or by developing alternative heuristics.

It is also important to emphasize that the network we obtained here cannot and should not be used for large scale biological inference, because the structure of the network is strongly influenced by research biases and the relationship with the true regulatory network is very uncertain. The highly skewed TF coverage among the candidate papers, coupled with the correlation between the number of candidate papers and gene popularity implies that researchers generally choose to study DTRIs involving TFs of previously known significance. Conversely, some genes, such as TBR1, are functionally important but lack experimental characterization, perhaps due to their more recently discovered functional roles. This general research bias, combined with biases of our curation, has obvious impacts on the resulting network structure. Previous work by our group has documented the impact of bias towards well studied, multifunctional genes in other types of network analyses [43]. Our observation of the high internode connectivity in the curated network demonstrates the presence of DTRI biases beyond gene popularity. Likewise, it is unclear whether the skewed representation of DTRIs involving activation of proximal cREs is the result of research bias or a real biological pattern. As such, we caution against interpretations based on the properties of the manually curated network.

We curated what we estimate to be a substantial, but still small, fraction of the relevant literature. Fortunately, our curation protocol can be scaled up to produce a considerably larger collection of high confidence DTRIs. According to our estimates, our current curation has captured less than ten percent of all experimentally verified DTRIs reported in the published literature. The bulk of our curation was performed in four months by two full time curators. Given this experience, we estimate an exhaustive curation effort could be completed by a team

of ten curators in approximately 12 months. Importantly, we predict that about a third of all reported DTRIs would be supported by all three types of experimental evidence. However, we take note of the scarcity of specific DTRIs in particular contexts. In particular, we found less than 5% of all recorded DTRIs to have reliably demonstrated activity specifically in the CNS. While we postulate that a manual curation approach is required to establish a high confidence DTRI catalogue for training and validating high-throughput predictions, the aforementioned biases and scarcity of low-throughput experiments will prevent the use of manually curated networks directly for analysis. To elucidate the architecture of gene regulation underpinning neurodevelopment and disease, it is imperative to develop effective means for accurately predicting DTRIs based on high-throughput data. This curation effort supports progress towards this end.

## Methods

All data analyses and visualizations were performed using R and R-Studio [44]. Data plots were made using ggplot2 [45]. Networks visualizations were constructed using Cytoscape [46]. For data manipulation, we used Tidyverse [47].

### Obtaining records from external resources

We obtained records from eight external databases: ENdb, TRRUST, CytReg, OReganno, HTRIdb, TFe, TFactS, and InnateDB. We downloaded the ENdb records from http://www.licpathway.net/ENdb/ on Sept. 14th, 2020. Records from the remaining databases were downloaded between Dec. 9 and Dec. 16, 2019. We obtained the CytReg records from the supplementary data of the original publication. For TRRUST, we downloaded both the human and mouse data tables directly from https://www.grnpedia.org/trrust/. An additional column was added to preserve the species annotation before joining the two tables. The most recent version of the records in ORegAnno were obtained from http://www.oreganno.org/. Here, we retained only records with valid Entrez or Ensembl ID, and PubMed ID annotations. In addition, records annotated as miRNA regulation or those resulting from high-throughput screens were excluded. We downloaded InnateDB records from https://www.innatedb.com/ and filtered for records reporting protein-DNA interactions. The TFe records were retrieved from the now deprecated web API, http://cisreg.cmmt.ubc.ca/cgi-bin/tfe. Species information was inferred from the TF gene symbols recorded in TFe. The TFactS records were downloaded from http://www.tfacts.org/. A union set was derived by merging both signed and signless data tables in TFactS. Finally, for HTRIdb, we downloaded the data from http://www.lbbc.ibb.unesp.br/htri. Here, we filtered for literature curated records with valid PubMed ID annotations.

From each database we retained records of one-to-one regulator-target interactions with annotations in either human or mouse. We indexed genes using Entrez IDs. In cases where only the gene symbols were available, we mapped the symbols to Entrez IDs, first by using the official HGNC or MGI symbols and then by gene aliases. With the exception of ENdb, which was published after we completed curation, the retrieved set of publications was used as a source of candidate papers for curation in the present study (S1 Data). Each publication was assigned to one or more TFs based on the recorded DTRIs. Additionally, we also retained species information and modes of regulation where available. For all analysis and reporting, we matched all TF and target genes to human orthologs using NCBI HomoloGene [48] (Mancarci and French, 2019: https://cran.r-project.org/web/packages/homologene/index.html). When there are no human orthologs, the mouse Entrez gene is used directly. Independent regulatory interactions were defined as unique combinations of the human TF and target genes.

### Identification of neurodevelopmental TFs

We define TFs to be either the genes annotated with least one regulatory target in any of the previous resources or those identified as TFs by Lambert et al. (2018) [49]. Collectively, this TF set consists of 2,235 genes. Given our particular focus in this study on neurodevelopment, we further designated 438 TFs as neurodevelopmental TFs based Gene Ontology annotations, and disease association records from SFARI (S2 Data) [50,51]. We downloaded the list of genes annotated with the central nervous system development GO term (GO:0007417) or any of its descendent terms for both human and mouse from AmiGO (http://amigo.geneontology.org/). Next, we downloaded the list of genes associated with neurodevelopmental disorders from the SFARI database (https://gene.sfari.org/). The list of TFs is provided in S2 Data. Finally, we manually prioritized these TFs for curation based on the annotated association with neurodevelopment and the number of candidate papers retrieved.

### Obtaining candidate publications for curation

In addition to the candidate papers derived from the external resources, we also performed an independent PubMed query for each TF (refer to the previous section for the operational definition of a TF). We took advantage of the E-Utilities API provided by NCBI to perform searches programmatically [52]. We selected six MeSH terms that indicate experimental evidence for: "Regulatory Sequences, Nucleic Acid", "Transcription, Genetic", "Intracellular Signaling Peptides and Proteins", "Gene Expression Regulation", "Chromatin Immunoprecipitation", and "Electrophoretic Mobility Shift Assay". The set of the selected search terms were appended to the gene symbol of each TF to form an independent search query to obtain the corresponding set of candidate papers. To approximate gene popularity of TFs, we performed another round of PubMed query for each TF using only the gene symbol without the MeSH terms.

### Experiment-centric curation of DTRIs

For each paper that we examined, we recorded all low-throughput experimental evidence of DTRIs. Specifically, we look for three types of experiments: TF perturbation, TF-DNA binding, and TF-reporter assays. As such, each experiment constitutes an independent record in the database and is assigned a unique identifier (S3 Data). Gene identifiers were translated into Entrez IDs at the time of recording. Species information was recorded separately for the TF and the target genes. The context type may be cell lines, or primary tissue or cells. In the case of EMSA experiments, the context types are designated to be in-vitro. In addition to the context type, we further annotated each experiment with a specific ontology term in order to retain the highest context resolution possible. We used terms from the UBERON ontology [53] for primary tissue, the CL ontology [54] for primary cells, and the CLO ontology [55] for cell lines. Where the appropriate ontology term could not be found in the aforementioned ontologies, we additionally used terms from the BTO [56] and the EFO [57] ontologies. When all else fails, we directly recorded the name provided in the original publication. Age was also recorded as a separate attribute for experiments that used primary tissues or cells. Where available, we also recorded the direction of regulation as well as whether the reported regulatory element is proximal or distal to the TSS of the target gene. Proximal elements were defined to be either promoters or cREs within 2 kb upstream or downstream of the TSS. Box 1 contains the full list of recorded attributes and the corresponding descriptions.

For each type of experiment, we selected a number of details. For TF perturbation experiments, we recorded whether the TF was overexpressed, down regulated, or knocked out. We also recorded whether the perturbation was dynamically induced before the time of assay or

constitutively modified at the beginning of life. For TF-DNA binding experiments, we recorded both ChIP-assay and EMSA experiments. For EMSA, we further annotated the source of the TF protein. Finally, for reporter assays, we recorded whether mutations were introduced to the cRE sequence for comparison.

## Network analysis

To assess the connectivity of the manually curated network, we used the iGraph package in R [58]. First, we constructed a directed network consisting of all curated DTRIs. Three metrics were computed to measure internode connectivity: the number of valid gene-to-gene paths, the number of cliques with three or more nodes, and the number of independent components. To assess statistical significance, we constructed 1000 network permutations by randomly swapping all edges while preserving both in and out degrees of all nodes. This set of random networks were then used to generate empirical null distributions for each of the three metrics. One-tailed p-values were computed by obtaining the fraction of random values larger or smaller than the observed values.

## Comparison with high-throughput TF perturbation screens

We selected PAX6/Pax6, the TF with the highest number of recorded targets, as the primary TF for assessing correspondence with high-throughput screens. We obtained the genome wide expression data generated by Walcher et al. (2013) [36] along with its metadata from Gemma [59]. We then performed a differential expression analysis between the wild type and the Pax6-Sey samples using limma [60]. This resulted in a list of genes with p-values and FDR corrected q-values representing significance of differential expression upon Pax6 knockout. For hit list analyses, we used a cut-off FDR of 0.1. Processed differential expression analysis results generated by Narayanan et al. (2018) [37] were obtained directly from the supplementary materials of the original publication [37]. These two datasets were selected for their relevance to brain development. We used the ranking of nominal p-values for the AUROC. The Entrez IDs for the mouse genes were mapped to human orthologs using HomoloGene so that the results could be compared with the current curation.

   Next, we took all curated targets for PAX6/Pax6 that were present in each dataset and sliced it according to cellular context and quality of evidence. To retrieve targets with demonstrated activity in the CNS, we retrieved all interactions for PAX6/Pax6 where there is at least one experiment annotated with the CNS ontology term (UBERON:0001017) or any of its descendent terms. Similarly, we searched for all targets annotated with the eye term (UBERON:0000970). Targets with evidence in both the CNS and the eye were placed only in the CNS category so that the categories are mutually exclusive. The remaining targets were classified as "other". To subset by quality of evidence, we binned all PAX6/Pax6 targets into those with multiple types of experiments vs. only a single type of experiment.

   Each of these target subsets were then tested for enrichment in the two high-throughput differential expression screens. Enrichment was tested in two ways. First, for Walcher et al., 2013 [36], a hit list of differentially expressed genes was generated using an FDR threshold. Overrepresentation of the curated targets in this list was tested by using the hypergeometric distribution, yielding a p-value for each set of curated targets. Next, we generated a ranking of differentially expressed genes in both datasets using nominal p-values and used the AUROC method to test enrichment for each set of curated targets towards the top of these rankings. AUROCs were computed by using the Mann-Whitney U Test. To test for significant differences among the AUROCs for the different target subsets, we estimated the variance of the AUROC for each subset by bootstrapping 1000 random samples for each set of curated targets.

Significant differences were determined by assessing the overlaps among 95th percentile confidence intervals.

Beyond PAX6/Pax6, we additionally compared the curated targets for other TFs to high-throughput screens sourced from KnockTF. The complete KnockTF database was downloaded from http://www.licpathway.net/KnockTF/ on June 6, 2021. Initially, we included all the TFs with 10 or more curated targets in the current resource and at least three TF perturbation screens (S13 Data). Across the 10 TFs, we collectively obtained the processed differential expression analysis results for 53 datasets. Like the PAX6/Pax6 datasets, we used the rankings of nominal p-values to perform the AUROC enrichment analyses. To control for dataset specific conditions, we obtained the consensus ranking for each TF by averaging the rank of each gene across the datasets. Nonoverlapping genes were dropped. To compare across all DTRI resources, we additionally obtained the results of 123 TF perturbation datasets, for a total of 176 datasets and 37 TFs. Using the consensus rankings, AUROCs and corresponding p-values were then computed for all DTRI resource and TF combinations with at least 10 recorded targets.

## Supporting information

**S1 Fig. Distribution of TFs by the number of targets.** Only TFs with at least one curated target are plotted. Most (212/251) TFs have less than ten targets recorded.
(PDF)

**S2 Fig. Distribution of targets by the number of recorded TF regulators.** Only targets with at least one recorded TF regulator are included.
(PDF)

**S3 Fig. Number of TF regulators per target for the top 50 target genes.** Target genes that are also neurodevelopmental TFs are indicated by blue dots. Bar colors correspond to cellular contexts indicated in the color legend. There are eight targets with more than 10 recorded TF regulators (denoted by the horizontal dotted line).
(PDF)

**S4 Fig. Details of TF perturbation experiments.** Colors correspond to cellular contexts as indicated in the legend. Top left: Breakdown of experiments by species. The "mixed" column refers to experiments where the TF and the target genes originated from different species, which is possible in TF overexpression experiments. Top right: Breakdown of experiments by context type. Bottom left: Breakdown of experiments by the mode of TF perturbation. "Knock Out" refers perturbations at the genetic level including naturally occurring mutations. Further breakdown into heterozygous or homozygous knock outs are provided in S5 Data. "Knock Down" refers to transcript level perturbation by RNA interference. Bottom right: Breakdown of experiments by the effect of TF perturbation in primary tissues or cells. "Constitutive" perturbations are present throughout development versus "induced" perturbations are triggered closer to the time of assay.
(PDF)

**S5 Fig. Details of TF-DNA binding experiments.** Colors correspond to cellular contexts as indicated in the color legend. Top left: Breakdown of experiments by species. The "mixed" column refers to experiments where the TF and the target genes originated from different species, which is possible in EMSA experiments. Top right: Breakdown of experiments by context type. All EMSA experiments were classified as "in-vitro" for context type. Bottom left: Breakdown of experiments by method. Bottom right: Breakdown of experiments by source of the TF protein in EMSA experiments. Forty-eight EMSA experiments used endogenous TF proteins

sourced from primary tissues or cells. In most cases (292), TF proteins were sourced from cell lines following TF transfection.
(PDF)

**S6 Fig. Details of TF-reporter experiments.** Colors correspond to cellular contexts as indicated in the color legend. Top left: Breakdown of experiments by species. The "mixed" column refers to experiments where the TF and the target genes originated from different species. Top right: Breakdown of experiments by context type. Few TF-reporter experiments were performed using primary tissues or cells. Bottom left: Breakdown of TF-reporter experiments that did or did not investigate the effect of mutating the corresponding cRE sequence. Bottom right: Breakdown of whether the effect of the mutation on the TF-DNA interaction was confirmed experimentally by EMSA.
(PDF)

**S7 Fig. Breakdown of experiments performed using primary tissues or cells across anatomical systems.** DTRI counts are plotted on the left and experiment counts are plotted on the right. Additional columns are shown for experiments performed using embryonic tissue or cells. DTRIs and experiments involving neurodevelopmental TF regulators are highlighted in blue.
(PDF)

**S8 Fig. Breakdown of experiments performed using cell lines across broad cell type categories.** DTRI counts are plotted on the left and experiment counts are plotted on the right. DTRIs involving neurodevelopmental TF regulators are highlighted in blue.
(PDF)

**S9 Fig. Pairwise overlap of DTRIs among the different data resources.** Overlap values are reported as fractions of the "target" resource (x-axis). Only values of 0.05 or higher are printed. For example, TRRUST contains 0.4 of the DTRIs recorded in our curation whereas we captured less than 0.05 of the DTRIs in TRRUST. External resources are ordered by the number of recorded DTRIs.
(PDF)

**S10 Fig. Distribution of papers by the number of DTRIs reported.** Only papers with at least one curated DTRI are included. The majority of papers report only a single DTRI.
(PDF)

**S11 Fig. Breakdown of DTRIs by mode of regulation and TFBS position.** For "TFBS Position" (y-axis), "Proximal" refers to promoters or cREs that are within 2 kb of the target TSS as reported in the original publication. The "Distal" category includes cREs further than 2 kb upstream or downstream from the target TSS. "Both" includes DTRIs with both proximal and distal regulatory elements. For "Mode of Regulation", "Activation" refers to TF perturbation or TF-reporter assays where the direction of change in target gene expression is the same as direction of the corresponding TF perturbation. Similarly, "Repression" includes results where expression of the target gene changes in the opposite direction. "Both" refers to DTRIs with alternative results from different experiments.
(PDF)

**S12 Fig. TFBS Position of DTRIs per TF for the top 50 TFs.** Colors correspond to TFBS position annotations. Dotted line indicates 10 targets on the y-axis for reference.
(PDF)

**S13 Fig. Distribution of TFBS coordinates across experiments.** TFBS coordinates are given relative to the TSS of the target gene. Only the coordinates of the ends closer to the target genes' TSS were recorded. Dotted lines (+/-2000) indicate the threshold for calling proximal or distal TFBSs. Records with coordinates above or below +/-10000 are not included in this figure.
(PDF)

**S14 Fig. Mode of regulation of DTRIs per TF for the top 50 TFs.** Colors correspond to the mode of regulation annotations. The dotted line indicates 10 on the y-axis for reference. Most TFs appear to be primarily activators whereas a small number of TFs such as HES1, GLI3, and REST repress the majority of their recorded targets.
(PDF)

**S15 Fig. Enrichment of all curated PAX6/Pax6 targets among differentially expressed genes in Walcher et al., 2013 [36].** AUROC and the corresponding p-value (Mann-Whitney U Test) are displayed in the panel.
(PDF)

**S16 Fig. Enrichment of curated PAX6/Pax6 targets among differentially expressed genes in Walcher et al., 2013 [36] by tissue types.** Targets are classified based on the reported cellular contexts. AUROCs and the corresponding p-values (Mann-Whitney U Test) are displayed in the panel. Color coding corresponds to categories of targets. Targets with experimental validation in primary CNS tissues are most enriched in the perturbation screen. No statistically significant enrichment is observed for targets tested in tissues or cell types other than the CNS or the eye.
(PDF)

**S17 Fig. Enrichment levels (measured in AUROC) for the different categories of curated PAX6/Pax6 targets among differentially expressed in genes in Walcher et al., 2013 [36].** Confidence intervals (95th percentile) were derived by bootstrapping 1000 random samples from each category. Statistically significant differences were observed between the CNS versus "other". A large difference was also observed between targets with multiple versus a single type of low-throughput experimental evidence, though it did not pass the threshold for statistical significance. Dotted line of AUROC = 0.5 indicates random expectation.
(PDF)

**S18 Fig. Enrichment of curated PAX6/Pax6 targets among differentially expressed genes in Walcher et al., 2013 [36] by number of experiment types.** AUROCs and the corresponding p-values (Mann-Whitney U Test) are displayed in the panel. Color coding corresponds to categories of targets. No statistically significant enrichment is observed for targets tested only in a single type of experiment.
(PDF)

**S19 Fig. Enrichment levels (measured in AUROC) for the different categories of curated PAX6/Pax6 targets among differentially expressed in genes in Narayanan et al., 2018 [37].** Confidence intervals (95th percentile) were derived by bootstrapping 1000 random samples from each category. The general trend from the comparison with Walcher et al., 2013 [36] was replicated (S17 Fig) though the differences did not pass the threshold for statistical significance here. Dotted line of AUROC = 0.5 indicates random expectation.
(PDF)

**S20 Fig. Enrichment of curated PAX6/Pax6 targets among differentially expressed genes in Narayanan et al., 2018 [37] by tissue types.** AUROCs and the corresponding p-values (Mann-Whitney U Test) are displayed in the panel. Color coding corresponds to categories of targets. Only the targets experimentally verified in the CNS show statistically significant enrichment.
(PDF)

**S21 Fig. Enrichment of curated targets in TF perturbation screens sourced from KnockTF.** Each point is a high-throughput data set. Stars denote enrichment in the consensus ranking across all datasets for each TF. TFs are ordered by decreasing AUROCs in the consensus ranking. For eight of the 10 TFs, the consensus rankings are more enriched for the curated targets than the median across all datasets. Colors indicate statistical significance. For four TFs, the consensus rankings are significantly enriched for the curated targets.
(PDF)

**S22 Fig. Comparison of enrichment levels among DTRI resources.** AUROC enrichment values were computed for each unique combination of DTRI resource and TF where there 10 or more targets using the consensus rankings (See Methods). The size of each data point corresponds to the number of TFs where the AUROC values were computed. The data points show average AUROCs, and the error bars show the 95th confidence intervals. A confidence interval could not be computed for ENdb as this resource only contains enough (>10) targets for a single TF.
(PDF)

**S1 Data. List of candidate papers.**
(TSV)

**S2 Data. List of TFs.**
(TSV)

**S3 Data. Curated records at the experiment level.**
(TSV)

**S4 Data. Curated records summarized at the DTRI level.**
(TSV)

**S5 Data. TF Perturbation experiment details.**
(TSV)

**S6 Data. TF-DNA Binding experiment details.**
(TSV)

**S7 Data. TF-reporter experiment details.**
(TSV)

**S8 Data. All DTRI records including those curated in external databases.**
(TSV)

**S9 Data. Records obtained from external databases.**
(TSV)

**S10 Data. Reprocessed Pax6 knockout differential expression analysis results using data from Walcher et al., 2013 [36].**
(TSV)

**S11 Data. Experiments with negative results.**
(TSV)

**S12 Data. Curated records annotated with the PSI-MI vocabulary.**
(TSV)

**S13 Data. Number of targets per TF across DTRI resources.**
(TSV)

**S1 Appendix. Curation Protocol.**
(PDF)

## Acknowledgments

We thank Dr. Shamsuddin Bhuiyan for providing advice on setting up the initial curation protocol. We thank Nathaniel Lim for providing the formatted ontology hierarchy files for tissue and cell type analyses. We thank Dr. Shamsuddin Bhuiyan, Dr. Marjan Farahbod, and Dr. Sanja Rogic for advice on writing and providing feedback on an early draft.

## Author Contributions

**Conceptualization:** Eric Ching-Pan Chu, Alexander Morin, Paul Pavlidis.

**Data curation:** Tak Hou Calvin Chang, Tue Nguyen, Yi-Cheng Tsai, Aman Sharma, Chao Chun Liu.

**Formal analysis:** Eric Ching-Pan Chu.

**Funding acquisition:** Paul Pavlidis.

**Methodology:** Eric Ching-Pan Chu, Alexander Morin, Paul Pavlidis.

**Supervision:** Paul Pavlidis.

**Visualization:** Eric Ching-Pan Chu.

**Writing – original draft:** Eric Ching-Pan Chu.

**Writing – review & editing:** Alexander Morin, Paul Pavlidis.

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
