## [Decision Letter · Decision Letter 0]

4 Jun 2021

Dear Dr Pavlidis,

Thank you very much for submitting your manuscript "Experiment level curation identifies high confidence transcriptional regulatory interactions in neurodevelopment" for consideration at PLOS Computational Biology.

As with all papers reviewed by the journal, your manuscript was reviewed by members of the editorial board and by two independent reviewers. In light of the reviews (below this email), we would like to invite the resubmission of a significantly-revised version that takes into account the reviewers' comments.

The reviewers' suggestions for improving the study are quite clear: I would like to particularly emphasise the need for additional validation (both reviewers) and the availability of the protocols and data in standardised formats (reviewer 2).

We cannot make any decision about publication until we have seen the revised manuscript and your response to the reviewers' comments. Your revised manuscript is also likely to be sent to reviewers for further evaluation.

Sincerely,

Kiran Raosaheb Patil, Ph.D.

Deputy Editor

PLOS Computational Biology

Ilya Ioshikhes

Deputy Editor

PLOS Computational Biology

The reviewers' suggestions for improving the study are quite clear: I would like to particularly emphasise the need for additional validation (both reviewers) and the availability of the protocols and data in standardised formats (reviewer 1).

Reviewer's Responses to Questions

**Comments to the Authors:**

Reviewer #1: In this manuscript the authors describe the construction of a manually curated collection of TF-gene regulatory interactions, involving 251 TFs associated with nervous system development. Although being restricted to a small set of TFs of interest, this work provides a valuable resource for the community, containing important information such as type of experimental evidences of the TF-gene interactions and cellular context, adding to the novelty of the described database compared to the published resources. The analyses are generally well performed, the curation protocol is well described and all data and source code is made available. Overall, this study merits publication in PLOS Computational Biology. I have however some comments that I believe need to be addressed.

1) Authors validate their database using one example of a Pax6 perturbation study. To strengthen their approach, they should consider providing 1-2 more examples of other TFs for which there is available perturbation data.

2) To compare the performance of their approach with the available databases, the authors should provide Pax6 targets enrichment results using 1-2 other published resources they mention, e.g. TRRUST or CysReg.

3) Minor comment: Figures 2A and 4D – colours of the Venn diagrams are too dim, making the diagrams not visible.

Reviewer #2: The reviewer thanks the authors for the opportunity to gain insight in their interesting work and thanks the PLOS Computational Biology for the opportunity to act as reviewer.

The availability of high quality curated regulatory information annotated with evidence from low-throughput experiments and information regarding mode of regulation and biological context is of high importance for all aspects of life sciences that employ computational approaches involving comprehensive querying and modelling of digital standardised data, information and knowledge with the aim to seek improved understanding of complex biological systems to optimally support research and innovation for societal sectors, including health care, food, biotech industry and environment. Biocuration, as accomplished in the current work - the extraction of knowledge from unstructured biological data into a structured, computable form – is scientific work of high importance and impact. The large set of information regarding transcriptional regulatory interactions involving transcription factors with roles in neurodevelopment generated by the authors represents a highly valuable resource, in particular for the field of neurobiology. Specifically, the curated compendium of information is of commendable richness in experimental and biological context detail, all of which is systematically mapped to acknowledged ontologies and controlled vocabularies. Moreover, the paper includes analyses of the curated information allowing for prudent and informative discussion of research bias embodied in the scientific literature and challenges related to lack of completeness of information on transcriptional regulatory interactions in existing databases.

The authors hypothesize that information regarding direct transcriptional regulatory interactions (DTRIs) evidenced by low-throughput experiments as documented by curation of scientific literature, can produce a reliable set of DTRIs that is highly interpretable and more suitable for the evaluation of high-throughput predictions than other, similar resources . The group conducted a curation effort to generate a set of 1499 DTRIs focussing 251 transcription factors (TFs) considered to be involved in neurodevelopment, and for which evidence from low-throughput experiments could be found in the scientific literature. The hypothesis was then tested by comparing the DTRIs in the curated set with the content in existing resources and with transcriptional regulatory interactions obseved in a previousy published high-throughput Pax6 perturbation screen. Based on finding an enrichment of curated, experimentally evidenced Pax6 DTRIs among the transcriptional interactions detected in the high-throughput screen, the authors claim that the literature curated set of DTRIs represents a reliable knowledge resource with high sensitivity for DTRIs predicted from HTP screens and more suitable for the evaluation of high-throughput predictions than other, similar resources. Furthermore, the authors claim that their curated set af DTRIs is more interpretable than other similar resources.

The originality of the claims is modest, since it is widely accepted that information and knowledge manually curated from full text papers where it is normally supported by low-throughput experimental evidence has high value for the current life sciences. This has been widely discussed and documented, amongst others through the activites of the Gene Ontology Consortium and IMEX Consortium.

Specifically, the value of such information and knowledge in the field of transcriptional regulatory interactions, has been convincingly demonstrated by Garcia-Alonso L, et al, 2019 (PMID: 31340985). An in depth manual curation effort for transcriptional regulatory interactions within the subdomain of liver-enriched TFs by e.g. Thomas P, et al, 2015 (PMID: 25433699) similarly demonstrated such value.

The manuscript’s support for the claims that the curated set of DTRIs is highly reliable and is more interpretable and more suitable than other, similar resources for the evaluation of predictions from high-throughput experiments is on the weak side. This is related to the circumstance that set of DTRIs has only been tested against one HTP screen probing TF Pax6 transcriptional regulations and that the attribute ‘interpretability’ has not been rigorously defined and tested.

The reviewer suggests revision of the manuscript comprising, amongst other, more coherent articulation of and stronger support for the central claims to make a stronger case for publication as research paper in PLOS Computational Biology

Areas of improvement

1) The curated set of DTRIs should be tested against additonal HTP screens representing a wider repertoir of TFs, and ideally including more than one type of HTP analysis (for instance ChIP in addition to transcriptomics) to derive more robust support for the claim pertaining to suitability for evaluation of high-throughput predictions.

2) To support the claim pertaining to higher interpretability, it would be necessary for the authors to define what they mean by interpretability. This should be followed by testing against other, similar resources.

3) Of central focus of the presented work is the curation of «direct transcriptional regulatory interactions (DTRIs)». However, the authors do not specify what they mean by «direct». According to established conventions, the regulators involved in «direct» transcription regulation are regarded to be the DNA -binding transcription factors (dbTFs) whereas the transcription co-regulators are considered to be more indirectly involved via protein interactions with dbTFs. GO-terms GO:0003700 and GO:0003712 can be consulted for authoritative definitions of the dstinct molecular functions of dbTFs and co-TFs. The TFs in the curated DTRI set comprises regulators of both categories. Examples of proteins in the set that are considered to be co-TFs and not dbTFs are CTNNB1 and SPOP. The reviewer is aware that term «transcription factor» is often used for both dbTFs and co-TFs, also in several of the existing databases used and cited in the current work. Nevertheless, it is of high importance for use and interpretation of the valuable resource generated in the present work that it is explicitly stated what is meant by «direct», and to enable users of the data to adequately deal with dbTFs and co-TFs in the gene regulatory networks enabled by the resource.

4) Related to the issue discussed in 3), the authors should clarify the relevance and confidence of a direct regulatory interaction investigated by ChIP- and reporter experiments, compared to experiments like EMSA performed with purified TFs. The reviewer regards the latter as a clearly stronger evidence for direct protein-gene interaction than the former, even though some reporter gene experimental set ups, e.g. those utilizing gene regulatory regions specifically mutated in the TFBS, can also provide strong indication for direct interaction. The reviewer disagrees with the authors that «…reliability of EMSAs might be improved by using the endogenous TF protein…» in nuclear extracts (manuscript line 232-). It is commonly thought that it is not possible to ascertain the nature of the protein (complex) represented in an EMSA band shift produced from nuclear extracts, meaning that it is not known which protein species are involved, and which specific protein(s) directly bind the DNA.

5) The manuscript refers to the curation protocol developed for this work. However the reviewer was unable to find this protocol in the submitted material. The protocol should be made available in a form that would enable a better understanding of how the curated information was identified and mapped to the chosen ontologies and controlled vocabularies and to enable other curators to reproduce the developed methodology for similar curation efforts. It would be highly preferable that the protocol includes guidelines pertaining to which sections of the full paper (e.g. result section, figures, method sections) to be used for the different curation aspects and how the identified information should be used to generate the appropriate annotations.

6) The definition of proximal and distal elements applied in the current work is not in line with the most updated definition for proximal-distal TFBS provided by the highly authoritative Sequence Ontology (SO). For example, the SO defines a proximal TFBS (SO:0001668) to be located in the region from about -250 to -40 relative to +1 of RNA transcription start site. The authors should consider to apply the definitions laborated by SO, or at least comment on how they deviate from this definition.

7) The data of the paper are made available in large detail and depth in supplementary files. However, in order to maximally enable the re-use of this valuable information, the authors are strongly adviced to re-check how the presentation of the curated data complies with established standards for representing regulatory interactions with the aim to implement adjustments that could enhance interoperability with other resources. Such standards have been formulated and discussed by e.g. the Gene Ontology Consortium, 2021 (PMID: 33290552), Perfetto et al, 2019 (PMID: 30793173) and Touré et al, 2021 (PMID: 33547799). The PSICQUIC exchange format developed by the EBI is well suited for dissemination of regulatory interaction information and could be considered for use by the authors in order to optimally support re-use and comply with FAIR principles.

Trondheim June 3, 2020

Yours sincerely

Astrid Lægreid, PhD

Professor Functional Genomics

Department of clinical and molecular medicine

Norwegian University of Science and Technology

Trondheim

Norway

**Have the authors made all data and (if applicable) computational code underlying the findings in their manuscript fully available?**

Reviewer #1: Yes

Reviewer #2: Yes

PLOS authors have the option to publish the peer review history of their article (what does this mean?). If published, this will include your full peer review and any attached files.

Reviewer #1: No

Reviewer #2: **Yes: **Astrid Lægreid
---

## [Decision Letter · Decision Letter 1]

20 Aug 2021

Dear Dr Pavlidis,

Thank you very much for submitting your manuscript "Experiment level curation identifies transcriptional regulatory interactions in neurodevelopment" for consideration at PLOS Computational Biology. As with all papers reviewed by the journal, your manuscript was reviewed by members of the editorial board and by several independent reviewers. The reviewers appreciated the attention to an important topic. Based on the reviews, we are likely to accept this manuscript for publication, providing that you modify the manuscript according to the review recommendations.

Sincerely,

Kiran Raosaheb Patil, Ph.D.

Deputy Editor

PLOS Computational Biology

Ilya Ioshikhes

Deputy Editor

PLOS Computational Biology

[LINK]

Reviewer's Responses to Questions

**Comments to the Authors:**

Reviewer #1: The authors have addressed all of my comments very thoroughly and I would like to acknowledge them for their efforts. The main value of the work in the availability of experimental and contextual details of the curation is now clearly highlighted in the text and newly added validations and comparisons further strengthened the manuscript.

Reviewer #2: To the editor,

The reviewer thanks the journal for the opportunity to comment on the revised manuscript and thanks the authors for their constructive responses to the comments from reviewers

There are a few aspects where the manuscript would benefit from clarifications in the text. These are mentioned below in the order of their appearance in the text

*line 343*

It would be good if you refer to ENCODE nomenclature as basis for your definition of proximal (since there are a number of different definitions currently in use)

*line 366 and other similar parts of text*

In your letter to the editor on the revised manuscript you state that the comparison with the high throughput Pax6 perturbation was «intended to be a demonstration of a potential use case of our resource» (page 2).

In the manuscript your statements regarding the motivation and outcome of comparing the DTRI resource with observations in high-throughput screens are somewhat heterogenous and span statements pertaining to assessment of high-throughput screen results («...benchmark high-throughput screens» (line 366), «...high confidence DTRI catalogue for training and validating high-throughput predictions» (line 542)) but also pertaining to qualities related to your DTRI resource (e.g. «...suggest that our tissue annotations are accurate» (line 412))

It would be good if you could take another look to see whether clarity and stringency can be improved regarding motivations for comparison and premisses for statements regarding the outcome of the comparisons.

*line 418*

I suggest to provide a few words specifying biological context regarding the «alternative Pax6 pertubation dataset»

*line 424*

Is it 53 or 52 experiments? (Methods section states 52)

*line 424/line 689*

It should be stated whether «all the TFs with more than 10 curated targets» (line 689 Methods section) were taken from only the current DTRI resource, or from the compiled resource including all external resources. If it is the former (only current), this selection strategy is likely to create a bias in comparison with other DTRI resources because the set of TFs with at least 10 targets is likely to vary between resources.

*line 424/line 689*

It would be helpful for the reader if some details were given in the Result- or Methods section regarding the biological context of the 52/53 TF perturbation sets (cell line, in vivo, organ/tissue type, type of perturbation)

**Have the authors made all data and (if applicable) computational code underlying the findings in their manuscript fully available?**

Reviewer #1: Yes

Reviewer #2: Yes

PLOS authors have the option to publish the peer review history of their article (what does this mean?). If published, this will include your full peer review and any attached files.

Reviewer #1: No

Reviewer #2: **Yes: **Astrid Lægreid, PhD

Professor Functional Genomics

Department of Clinical and Molecular Medicine

Norwegian University of Science and Technology

Trondheim, Norway

Figure Files:

Data Requirements:

Reproducibility:

References:

---

## [Decision Letter · Decision Letter 2]

28 Sep 2021

Dear Dr Pavlidis,

We are pleased to inform you that your manuscript 'Experiment level curation of transcriptional regulatory interactions in neurodevelopment' has been provisionally accepted for publication in PLOS Computational Biology.

Best regards,

Kiran Raosaheb Patil, Ph.D.

Deputy Editor

PLOS Computational Biology

Ilya Ioshikhes

Deputy Editor

PLOS Computational Biology

Reviewer's Responses to Questions

**Comments to the Authors:**

Reviewer #2: The reviewer thanks the authors for their responses to comments

**Have the authors made all data and (if applicable) computational code underlying the findings in their manuscript fully available?**

Reviewer #2: Yes

PLOS authors have the option to publish the peer review history of their article (what does this mean?). If published, this will include your full peer review and any attached files.

Reviewer #2: **Yes: **Astrid Lægreid, PhD, Professor Functional Genomics; Department of Clinical and Molecular Medicine; Norwegian University of Science and Technology, Trondheim, Norway

---

## [Editor Report · Acceptance letter]

8 Oct 2021

PCOMPBIOL-D-21-00675R2 

Experiment level curation of transcriptional regulatory interactions in neurodevelopment

Dear Dr Pavlidis,

I am pleased to inform you that your manuscript has been formally accepted for publication in PLOS Computational Biology. Your manuscript is now with our production department and you will be notified of the publication date in due course.

With kind regards,

Olena Szabo
